# IReEn: Reverse-Engineering of Black-Box Functions via Iterative Neural Program Synthesis

**Hossein Hajipour**[1]**, Mateusz Malinowski** [2]**, Mario Fritz**[1]

[1]CISPA Helmholtz Center for Information Security
[2] DeepMind
{hossein.hajipour,fritz}@cispa.saarland,
mateuszm@google.com

## Abstract

In this work, we investigate the problem of revealing the functionality of a black-box agent. More specifically, we are interested in a formal description of the behavior of such an agent. This task is also known as *reverse engineering* and plays a pivotal role in software engineering, computer security, and most recently in explainability. In contrast to prior work, we do not rely on privileged information on the black-box, but rather investigate the problem under a weaker assumption of having only access to inputs and outputs of the agent. We approach this problem by iteratively refining a candidate set using a generative neural program synthesis approach until we arrive at a program that mimics the agent's behavior. We assess the performance of our approach on the Karel dataset. Our results show that the proposed approach even outperforms prior work that had privileged information on the black-box function.

Extended version available at `https://arxiv.org/abs/2006.10720` .

## 1 Introduction

Reverse-engineering is about gaining insights into the inner workings of a mechanism. In our work, we consider a program to be a black-box function that we can only interface with it through input-output interactions. This is a desired scenario in software engineering [9, 5], and security[8, 17]. Furthermore, a similar paradigm is used to seek interpretation for reinforcement learning agents [16]. Similar principles have also been applied to the machine learning domain to copy black-box models [15, 10, 11].

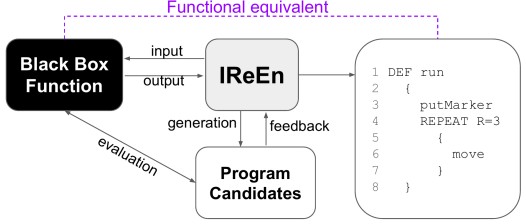

Figure 1: An example of revealing the functionality of a black-box function using only I/O interactions.

Despite all of the progress in reverse-engineering the software and machine learning models, there are typical limitations in the proposed works. E.g., in the decompilation task, one of the assumptions is to have access to the assembly code of the black-box programs, which is a significant information leak about the black-box function. Furthermore, in deep learning, a common issue is that the reverse-engineered models usually are not represented in an interpretable and human-readable form.

In recent work, neural program synthesizers are employed to recover a functional and interpretable form of a black-box program that is generated based only on I/Os examples [1]. On close inspection, however, it turns out that these approaches also leverage privileged information by relying on a biased sampling strategy of I/Os that was obtained under the knowledge of the black-box function.

In contrast to prior work, we propose an iterative neural program synthesis scheme to tackle the task of reveres-engineering in a black-box setting without any access to privileged information. Despite the weaker assumptions and hence the possibility to use our method broadly in other fields, we show that in many cases it is possible to reverse engineer programs on the Karel dataset. We even achieve

NeurIPS 2020 Workshop on Computer-Assisted Programming, Vancouver, Canada.

better results than prior work that has access to privileged information. We achieve this by an iterative program synthesis approach. We query a black-box function using random inputs to obtain a set of I/Os and refine the candidate set by an iterative neural program synthesis scheme. Note that iterative program synthesis was proposed before in [14, 6]. However, To the best of our knowledge, this is the first iterative neural program synthesis approach that operates in a black-box setting.

To summarize, the contributions of this work are as follow: 1. We propose an iterative neural program synthesis scheme to reverse-engineer the black-box programs. 2. We propose a functional equivalence metric in order to quantify progress on this task. 3. We evaluate our approach on Karel dataset, where our approach outperforms prior work despite having access to less information.

## 2 Problem Overview

In this section, we formulate the problem description. We base our notation on Bunel et al. [1].

**Program Synthesis.** Program synthesis deals with the problem of deriving a program in a specified programming language that satisfies the given specification. We treat input-output pairs $I/O = \left\{(I^k, O^k)\right\}_{k=1}^{K}$ as a form of specifying the functionality of the program. This problem can be formalized as finding a solution to the following optimization problem,

$$\underset{p \in \mathcal{P}}{\arg\min} \quad \Omega(p) \tag{1}$$

$$\text{s.t.} \quad p(I^k) = O^k \quad \forall k \in \{1, \dots, K\} \tag{2}$$

where $\mathcal{P}$ is the space of all possible programs written in the given language, and $\Omega$ is a measure for the programs. The situation is illustrated in Figure 2. As most program languages do not have a unique representation for a certain functionality. Naturally, by adding more constraints, we obtain a nested constraint set that converges towards the feasible set of functionally equivalent programs.

Program synthesis approaches usually use privileged information via biased sampling scheme in terms of specifications [1, 2, 12]. To arrive at the specifications, one has to know the functionality of the target program $P$ as they are designed to capture e.g. all branches of the program. We call these crafted specifications *crafted I/Os*.

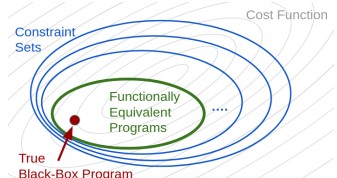

In this work, we focus on a black-box setting, where no such privileged information is available. Hence, we will have to randomly generate $K$ inputs $\{I^k\}_{k=1}^{K}$ and next query the program $p$ to obtain the corresponding outputs $\{O^k\}_{k=1}^{K}$. We call the obtained I/Os in the black-box setting *random I/Os*. It turns out (section 4), that indeed such random input queries yield significantly less information than the *crafted I/Os*.

Figure 2: Illustration of the optimization problem, functional equivalence, and feasible sets w.r.t. nested constraint sets.

## 3 Iterative Neural Program Synthesis

Reverse-engineering a black-box function and representing it in a high-level language is a challenging task. The main reason is that we can only interact with the black-box function using input-output examples. Also, solving the above constraint optimization problem is intractable. Therefore, in the following, we relax the optimization problem to a Bayesian inference problem and show how to iteratively incorporate additional constraints in order to synthesize the desired program.

### 3.1 Synthesizing Programs given Input-Output Constraints

To synthesize the program in black-box setting we optimize the neural program synthesizer $\Psi$ proposed by Bunel et al. [1]. This generative synthesizer outputs set of programs given $I/O$ constraints (Equation 3). In detail, we train an encoder-decoder model on a set of specifications $\{I/O_i\}_i$ and ground-truth programs $\{p_i\}_i$. Each specification is a set of $K$ pairs $I/O_i = \{(I_i^k, O_i^k)\}_{k=1}^{K}$ where the program needs to be consistent with I/Os, that is, $p_i(I_i^k) = O_i^k$ for all $k \in \{1, \dots, K\}$.

$$\hat{P} \sim \Psi(I/O). \tag{3}$$

Here, $\hat{P}$ is a set program candidate(s) $\{\hat{p}_1, ..., \hat{p}_C\} \in \hat{P}$ and $C \geq 1$. During inference, we condition synthesizer $\Psi$ on *random I/Os* to synthesize the underlying program of the target black-box function.

## 3.2 Sample Rejection Strategy

Naturally, we expect approximation errors of the optimization problem by the generative model. One main source of the error is that only a limited number of constraints can be incorporated in the model. To correct for these errors, we follow up with the sample rejection stage. We use random I/Os to evaluate the generated programs and score them based on the number of the I/Os which were covered by the programs. While in principle, any failed I/O should lead to rejecting a candidate, empirically, we find that keeping the highest scoring samples turns out to be advantageous.

## 3.3 Iterative Refinement

The generative model only takes a small number of constraints, while in black-box setting is unclear which constraints to use to arrive at the "functional equivalent" feasible set. Similar problems have been encountered in constraint optimization, where *delayed constraint generation* techniques have been employed to deal with a large number of constraints [4]. Motivated by these ideas, we propose an iterative strategy, where we condition in each step on a set of violated constraints that we find.

We present the details of the proposed approach in Algorithm 1. Itertive synthesis function takes synthesizer $\Psi$ and a set of I/Os of the black-box function (line 1). In line 2 we randomly choose a subset of I/Os as $I/O_s$. In line 3 we initial the $s_{best}$ to zero. Note that we use $p_{best}$ to store the best candidate, and $s_{best}$ to store the score of the best candidate. In the iterative loop, we condition the synthesizer on the given $I/O_s$ to get the program candidates $\hat{P}$ (line 6). Then we call *Scoring* function to score the program candidates in line 7. The scoring function returns the best candidate, the score of that candidate, and the new set of $I/O_s$, where the $I/O_s$ are the I/Os which were not satis-

---

**Algorithm 1:** Iterative Algorithms

---
1 **Function** IterativeSynthesis($\Psi$, $I/O$):
2     Initialize $I/O_s \subseteq I/O$.
3     $s_{best} = 0$ // To keep the best score.
4     n = *constant* // e.g. n=10
5     **for** $i \leftarrow 1$ **to** *n* **do**
6         $\hat{P} = \Psi(I/O_s)$
7         $\hat{p}_{best}, \hat{s}_{best}, I/O_s = Scoring(\hat{P}, I/O)$
8         **if** $s_{best} < \hat{s}_{best}$ **then**
9             $p_{best} = \hat{p}_{best}$
10             $s_{best} = \hat{s}_{best}$
11         **end**
12     **end**
13     **return** $p_{best}$
14 **End Function**

---

fied by $\hat{p}_{best}$. Here, $\hat{p}_{best}$, and $\hat{s}_{best}$ store the best candidate and the score of it for the current iteration. Then at line 8, we check if $\hat{s}_{best}$ is larger than the global score $s_{best}$, if the condition satisfies we update the global $p_{best}$, and $s_{best}$ (line 9-10). In line 13 we return the best candidate after *n* iterations.

# 4 Experiments

In this section, we show the effectiveness of our approach in synthesizing the black-box programs.

## 4.1 Dataset and Implementation Details

To evaluate our approach, we consider the Karel programming language. Karel featured a robot in a grid world, where this robot can move inside the grid world and modify the state of the world using a set of predefined functions and control flow structures. We use the Karel dataset published by Devlin et al. [3] to train and evaluate our models. In this dataset, for each program, there is 5 I/Os as specification, and one is the held-out for testing. This dataset contains 1,116,854 examples for training and 2,500 examples for testing the models. More details can be found in Appendix A.

To train the neural program synthesizer, we employ the neural networks architecture proposed by Bunel et al. [1], and we use that in our iterative approach as the synthesizer. We also fine-tune the pre-trained synthesizer on 100,000 pairs of random I/Os and target programs. To fine-tune the synthesizer, we use Adam optimizer [7]. We empirically find that fine-tuning the synthesizer can lead to better performance than training it using random I/Os. During inference, we use beam search with beam width 64 and select top-k programs. We provide more implementation details in Appendix B.

| Models | Generalization | | Functional | | Exact Match | |
|---|---|---|---|---|---|---|
| | top-1 | top-50 | top-1 | top-50 | top-1 | top-50 |
| Random I/Os | 57.12% | 71.48% | 49.36% | 63.72% | 34.96% | 40.92% |
| Random I/Os + FT | 64.72% | 77.64% | 55.64% | 70.12% | 39.44% | 45.4% |
| Random I/Os + IReEn | 76.20% | 85.28% | 61.64% | 73.24% | 40.95% | 44.99% |
| Random I/Os + FT + IReEn | **78.96%** | **88.39%** | **65.55%** | **78.08%** | **44.51%** | **48.11%** |
| Crafted I/Os (Bunel et al. [1]) | 73.12% | 86.28% | 55.04% | 68.72% | 40.08% | 43.08% |

Table 1: Top: Results of performance comparison of our approach in different settings using random I/Os for black-box program synthesis. Random I/Os mean that we use randomly obtained I/Os in the black-box setting, FT refers to fine-tuned model, and IReEn denotes to our iterative approach. Bottom: Results of Bunel et al. [1] when we use crafted I/Os. top-1 denotes the results for the most likely candidate, and top-50 denotes the results for 50 most likely candidates.

## 4.2 Functional Equivalence Metric

In [1, 13] two metrics have been used to evaluate the neural program synthesizer. 1. Exact Match: A predicted program is an exact match of the target if it is the same as the target program in terms of tokens. 2. Generalization: A predicted program is a generalization of the target if it satisfies specification and held-out I/Os. Both of these metrics have drawbacks. A predicted program might be functionally equivalent to the target but not be the exact match. On the other side, a program can be a generalization of the target by satisfying a small set of I/Os. However, it might not implement the functionality of the target. To overcome this issue, we propose the Functional Equivalence metric, where we consider a predicted program as an approximated functional equivalence of the target if it covers a large set of unseen I/Os. We provide more details of this metric in Appendix C.

## 4.3 Evaluation

To evaluate our approach in synthesizing the black-box programs, we query each black-box program with 50 valid random inputs to get the outputs. Note that we sample inputs from the same distribution that was used in [1]. In each iteration using the obtained 50 I/Os, we synthesize the programs, where we use 5 out of 50 I/Os to conditions on the synthesizer and use 50 I/Os to score the generated candidates. To evaluate the generated programs, in addition to generalization and exact match accuracy, we also consider the Functional Equivalence metric. To compute the approximated functional equivalency, we use 100 I/Os. If the generated program satisfies all of 100 I/Os we consider it as a program that is an approximately functional equivalent of the target program.

Table 1 shows the performance of our approach in different settings in the top, and the results of the synthesizer proposed by Bunel et al. [1] in the bottom. These results show that when we only use random I/Os (first row), there is a huge drop in the accuracy in all of the metrics in comparison to the results of the crafted I/Os. Based on these results we can see that random I/Os contain significantly less information about the target program in comparison to the crafted I/Os. However, when we use our iterative approach with the fine-tuned model (fourth row), we can see that our approach outperforms even the crafted I/Os in all of the metrics. Note that we apply our iterative synthesis for 10 iterations. We provide more evaluation results in Appendix D.

## 5 Conclusion

In this work, we propose an iterative neural program synthesis scheme to reverse-engineer the black-box functions. In contrast to previous works that have access to privileged information, we only rely on the input-output pairs. To tackle the problem of reverse-engineering the black-box function, we propose a neural program synthesizer in an iterative scheme to search for the best program candidate by conditioning the synthesizer on a set of violated constraints. Our evaluation on Karel dataset shows that our proposed approach even outperforms the previous work that uses privileged information.

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

# Appendix A    More Details of the Karel Task

We use the Karel dataset to evaluate our proposed approach. This dataset was created based on Karel programming language [12]. Figure 3 shows the grammar specification of this programming language [1]. In this grammar, we can see the details of the Karel programming language, including different statements, conditions, and actions. This figure shows that Karel consists of control flow structures such as conditionals and loops, which make this programming language a challenging and complex task. Figure 4 demonstrates an example of Karel task with two I/Os and the corresponding program.

To represent the Karel gird world (inputs and outputs), we use a tensor of size $16 \times H \times W$. Here we consider each grid world has a maximum size $16 \times 18 \times 18$. In this tensor representation, each cell in the grid is represented as a 16-dimensional vector. This 16-dimensional vector corresponds to the features described in Table 2.

$$
\begin{aligned}
\text{Prog } p \quad &:= \quad \texttt{def run() : } s \\
\text{Stmt } s \quad &:= \quad \texttt{while}(b) : s \mid \texttt{repeat}(r) : s \mid s_1; s_2 \mid a \\
&\quad \mid \quad \texttt{if}(b) : s \mid \texttt{ifelse}(b) : s_1 \texttt{ else } : s_2 \\
\text{Cond } b \quad &:= \quad \texttt{frontIsClear()} \mid \texttt{leftIsClear()} \mid \texttt{rightIsClear()} \\
&\quad \mid \quad \texttt{markersPresent()} \mid \texttt{noMarkersPresent()} \mid \texttt{not } b \\
\text{Action } a \quad &:= \quad \texttt{move()} \mid \texttt{turnRight()} \mid \texttt{turnLeft()} \\
&\quad \mid \quad \texttt{pickMarker()} \mid \texttt{putMarker()} \\
\text{Cste } r \quad &:= \quad 0 \mid 1 \mid \cdots \mid 19
\end{aligned}
$$

Figure 3: The grammar for the Karel programming language.

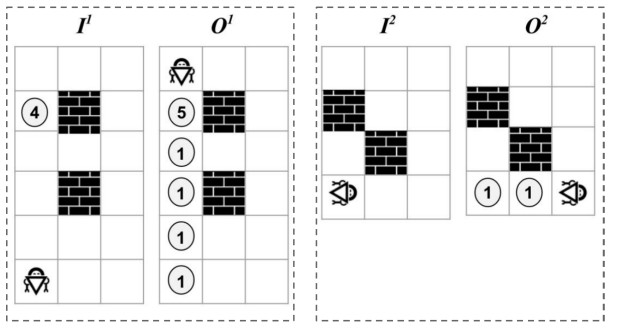

Figure 4: Example of two I/Os of a Karel task with the corresponding underlying program. The robot is Karel, the brick walls represent obstacles, and markers are represented with circles.

# Appendix B    Details of the Neural Program Synthesizer

One of the components of our iterative synthesizer scheme is the neural program synthesizer. To train this synthesizer, we follow the design of the proposed work by Bunel et al. [1]. Here, we train an encoder-decoder architecture to generate the programs given the input-output examples. In this section, we provide details of the neural network architecture and the training procedure of the program synthesizer.

## B.1    Neural Network Architecture

The encoder-decoder program synthesizer contains an encoder to encode the given I/O to a vector and a decoder that maps the given vector(s) to a set of program candidates. In particular, we use a convolutional neural network to encode the given I/O to a 512-dimensional vector. In the decoder, we have a 2-layer LSTM with a hidden size 256. Using this neural network architecture, the decoder generates the candidate programs given the encoded vectors of the I/Os.

| |
|---|
| Agent facing North |
| Agent facing South |
| Agent facing West |
| Agent facing East |
| Obstacle |
| Grid boundary |
| 1 marker |
| 2 marker |
| 3 marker |
| 4 marker |
| 5 marker |
| 6 marker |
| 7 marker |
| 8 marker |
| 9 marker |
| 10 marker |

Table 2: Grid representation of each cell in the Karel world.

## B.2 Training and Fine-tuning Details

To train the neural program synthesizer we use Adam optimizer [7], and the learning rate $10^{-4}$. We train the model for 100 epochs with a batch size of 128. Note that, to fine-tune the synthesizer model on random I/Os we use the learning rate $10^{-5}$, and we fine-tune the model for 10 epochs with a batch size of 128.

# Appendix C   Functional Equivalence Metric

In our work, we propose a new metric to evaluate the performance of the program synthesizers. We called this metric Functional Equivalence, where we consider the predicted program is approximately functional equivalent to the target program if it covers a large set of I/Os.

## C.1 Details and Parameter

To compute the results for approximated functional equivalency, we use 100 I/Os which were not seen by the program synthesizer. We get these I/Os by generating random inputs and query the ground truth program to get the corresponding outputs, and we check whether the random inputs hit all of the branches of the ground truth program. In our experiments, we found that with using more I/Os the more predicted programs we discover to not be functionally equivalent to the target programs. We found 100 number of I/Os as a point where the number of functionally equivalent programs stay stable in the evaluations.

## C.2 Examples of Functional Equivalence Programs

Here we provide examples of functional equivalent programs. We found these examples using the results of functional equivalence accuracy. In Figures 5 and 6 we provide two examples of functional equivalent programs, which are not exact match of each other. In these figures, the left box shows the target black-box program, and the right box shows the output program, which was predicted by our iterative program synthesizer approach. In Figure 5, we can see there is a difference between the target black-box program and the output program at lines 2 to 7. However, this difference is only a swap in the *ifelse* condition and the corresponding functions which were used in the *ifelse* statement, so it does not affect the functionality of the programs. Figure 6 show another example of functional equivalent programs. In this figure, we can see that in line 2 of the target program there is a *repeat* statement that was not used in the output program. However, when we turn an agent for four times, the agent's direction will remain the same, so turning the agent in the same place for five times is equal to turning the agent for one time, and we have *trungRight()* function in line 4 of the output

program, so having the *repeat* in target program does not affect the functional equivalency of these two programs.

**Target Black-Box Program**

```
1  def run():
2    ifelse(not(rightIsClear())):
3      turnLeft()
4      move()
5      pickMarker()
6    else:
7      turnRight()
8    move()
9    move()
10   pickMarker()
11   pickMarker()
```

**Output Program**

```
1  def run():
2    ifelse(rightIsClear()):
3      turnRight()
4    else:
5      turnLeft()
6      move()
7      pickMarker()
8    move()
9    move()
10   pickMarker()
11   pickMarker()
```

Figure 5: Example of functional equivalent programs. Left: Target black-box program from the test set. Right: Output of our iterative program synthesis approach.

**Target Black-Box Program**

```
1  def run():
2    repeat(5):
3      turnRight()
4    putMarker()
5    putMarker()
6    move()
```

**Output Program**

```
1  def run():
2    putMarker()
3    putMarker()
4    turnRight()
5    move()
```

Figure 6: Example of functional equivalent programs. Left: Target black-box program from the test set. Right: Output of our iterative program synthesis approach.

## Appendix D    More Analysis of Evaluation Results

Here, we investigate the effectiveness of the iterative refinement approach and also the effect of using the different numbers of I/Os for the sample rejection strategy.

### D.1    Effectiveness of Iterative Refinement

Figures 7a to 7c show the effectiveness of our proposed iterative approach in 10 iterations. In these figures, x and y axis refer to the number of iterations and the accuracy respectively. Figure 7a shows the exact match accuracy, Figure 7b shows the generalization accuracy, and in Figure 7c we can see the results of functional equivalence metric. In these figures, we provide results with and without fine-tuning the synthesizer. Here we can see the improvement of the exact match, generalization, and functional equivalence accuracy over the iterations. In Figure 7c we have a margin of 7% improvement in the functional equivalence accuracy for "Random I/Os + FT + IReEn" setting after 10 iterations. In other words, these results show that we can search for better program candidates by iteratively incorporate additional constraints.

### D.2    Effectiveness of Number of I/Os for Sample Rejection Strategy.

In our approach to choose one candidate among all of the generated program candidates, we consider a sample rejection strategy. To do that, we use the random I/Os to assign a score to the generated candidates based on the number of satisfied random I/Os. Finally, we consider the candidate with the highest score as the best candidate and reject the rest. Figures 8a to 8c show the effect of using the

different numbers of random I/Os on scoring the candidates and finding the best program candidate. x and y axis in these figures refer to the number of random I/Os and the accuracy of our approaches respectively. Figure 8a shows the exact match accuracy, Figure 8b shows the generalization accuracy and in Figure 8c we provide functional equivalence results. These figures show that by using more random I/Os in the scoring strategy, we can find more accurate programs that result to gain better performance. More specifically, by using more random I/Os for scoring the candidates, we can capture more details of the black-box function and find the best potential candidate among the generated one.

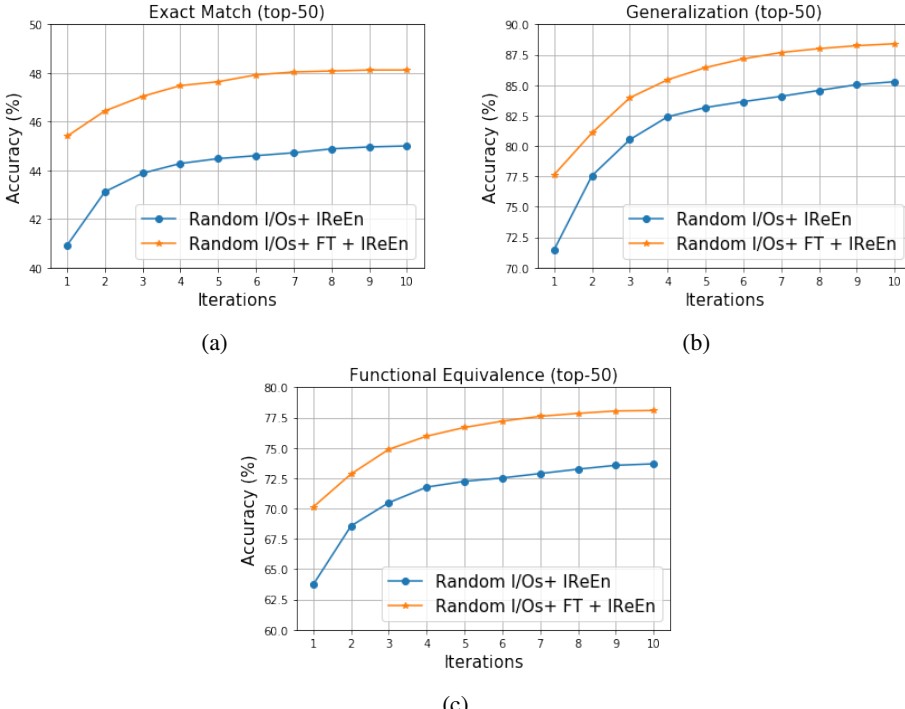

(a)                                                                                      (b)

(c)

Figure 7: (a) Exact Match accuracy after each iteration for "Random I/Os + IReEn" and "Random I/Os + FT + IReEn". (b) Generalization accuracy after each iteration for "Random I/Os + IReEn" and "Random I/Os + FT + IReEn". (c) Functional Equivalence accuracy after each iteration for "Random I/Os+ IReEn" and "Random I/Os + FT + IReEn". Note that, Random I/Os means that we use randomly obtained I/Os, FT denotes to the fine-tuned model, and IReEn refers to our iterative approach. Top-50 denotes the results for 50 most likely program candidates.

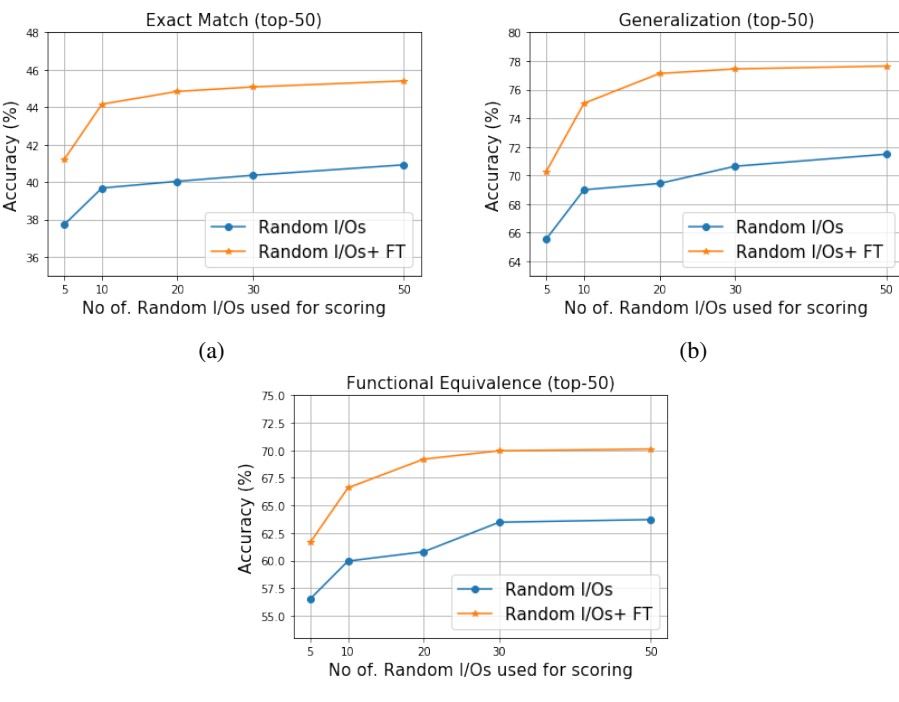

(a)                                              (b)

(c)

Figure 8: (a) Exact Match accuracy after using the different numbers of random I/Os in scoring strategy for "Random I/Os" and "Random I/Os + FT". (b) Generalization accuracy after using the different numbers of random I/Os in scoring strategy for "Random I/Os" and "Random I/Os + FT". (c) Functional Equivalence accuracy after using the different numbers of random I/Os in scoring strategy for "Random I/Os" and "Random I/Os + FT". Note that, Random I/Os means that we use randomly obtained I/Os, and FT denotes to the fine-tuned model. Top-50 denotes the results for 50 most likely program candidates.

