# OpenReview forum: "IReEn: Reverse-Engineering of Black-Box Functions via Iterative Neural Program Synthesis"
_NeurIPS.cc/2020/Workshop/CAP — NeurIPS 2020 CAP Workshop_

### Official Review · AnonReviewer1 · 2020-10-29
**Iterative Neural Program Synthesis in the Karel Domain**

**Rating:** 7
**Confidence:** 4

**Review:**

====== Summary ======
This paper presents a neural program synthesis approach for the Karel domain. While the technique only relies on input/output examples in order to synthesize programs, and could theoretically be used for other domains (i.e. the reverse engineering domain that is used to motivate the technique), it is unclear whether it would work in domains with very different forms of I/O. The technique uses a trained neural program synthesizer to output a set of candidate programs. To find the best candidate program, the technique repeatedly applies this synthesizer on subsets of random I/O examples, and keeps the candidate program that generalizes the best over all I/O examples. The approach is evaluated on the Karel dataset, and they find that the iterative approach, along with fine tuning, helps improve the performance of neural program synthesis.

====== Review ======
While this paper introduces the problem in the context of reverse engineering, it is only evaluated in the Karel domain. However, it considers the problem of program synthesis in this domain when each problem is specified as a set of _random_ input-output examples, as opposed to assuming there is a particularly well-crafted set of input-output examples. This restriction is motivated by the reverse engineering setting, but is also an interesting one to consider for program synthesis more broadly.

The core of the technique, as far as I understand it, uses several calls to a neural program synthesizer to generate candidate programs, and then filters out these candidates based on their generalization to held-out examples. I don't think anyone has tried this with a neural synthesizer as the generator of candidate programs in this domain. The idea of trying many different candidates and scoring them based on their performance on a held-out test set is in spirit similar CEGAR approaches to program synthesis, where an oracle tells the synthesizer if there is a an input-output pair that its proposed candidate does not satisfy. Here, we filter out the candidates instead of calling the synthesizer again.

The paper is fairly easy to read and understand. It could use a proof-read; I have pointed out some minor errors below.  I have outlined some key remaining question I have about the algorithm below, which should be clarified in the paper for future readers.

Nonetheless, it is interesting to see how the iterative approach can improve the results of the synthesis algorithm, almost totally making up for the negative effect that inferring on random input/output examples has (as opposed to inferring on carefully-crafted inputs). Seeing the difference in performance between crafted input output-examples and random ones is  The core idea of the paper is easy to understand, and I think this paper could spark discussion at the workshop. So I recommend it for acceptance.

====== Pros/Cons ======
Pros
+ Generalization/Function/Exact Match scores are higher than baseline
+ Ablation study
+ Shows the change in performance of prior work when I/O examples used for synthesis are not carefully crafted

Cons
- No information about the runtime of the approach, and some of the details
- Not a huge delta over prior work

====== Questions ======
- Why is the measure $\Omega$ introduced in equation (1)? It is never referred to in the rest of the paper as far as I can tell; I think the problem could be simplified to finding a program satisfying (2), no?
- Is the Sample Rejection Strategy (Section 3.2) just a subpart of Algorithm 1? My understanding is that Algorithm 1, Line 7, is exactly selecting the candidate that has the highest score on the random I/O examples. Am I missing something here? Where does 3.2 fit into the algorithm otherwise?
- Is the original synthesizer trained on the "hand-crafted" Karel dataset, and then only fine-tuned on Random I/Os? So the difference between Random and Crafter in Table 1 refers to the set of I/Os used at test time?
- In appendix C, it seems like Function Equivalence is very similar to Generalization. It might be good to contrast the two metrics here.

====== Minor ======
Line 5: "on" -> "of"?
Line 33: "reveres" -> "reverse"
Line 74: outputs ("a"?) set
Line 71: "model. one" -> "model. One"
Line 134: "equivalence" -> "equivalent"?

---

### Decision · Program_Chairs · 2020-11-02

**Decision:**

Accept

**Comment:**

The review is positive (and very thorough) so I recommend acceptance.